# Plain packaging of waterpipe tobacco? A qualitative analysis exploring waterpipe smokers' and non-smokers' responses to enhanced versus existing pictorial health warnings in Egypt

Aya Mostafa,[1] Heba Tallah Mohammed,[1,2] Wafaa Mohamed Hussein,[1] Mahmoud Elhabiby,[3] Wael Safwat,[4,5] Sahar Labib,[6] Aisha Aboul Fotouh,[1] Janet Hoek[7]

The abstract of this study has been presented at the World Conference on Tobacco or Health, 2018 in South Africa: https://doi.org/10.18332/tid/84640.

For numbered affiliations see end of article.

**Correspondence to**
Dr Aya Mostafa;
aya.kamaleldin@med.asu.edu.eg

## ABSTRACT

**Objective** Despite the global increase in waterpipe tobacco smoking (WTS) including in Egypt, few studies have assessed the effectiveness of waterpipe tobacco (WT) health warnings. Egypt has used pictorial health warnings (PHWs) on waterpipe tobacco packs (WTPs) and has rotated these every two years since 2008. We explored in this qualitative study how participants perceived existing PHWs on WTPs, assessed how they interpreted novel plain packaging of WT featuring enhanced PHWs, and probed perceptions of how existing and novel sets would affect uptake or cessation of WTS.

**Design** We conducted ten qualitative focus groups and ten in-depth interviews. We explored participants' views of the four existing PHWs (occupied 50% of the front and back of WTPs, displayed cancers, and featured colourful fruits and flavors) and four novel PHWs (occupied 80% of the front and back of WTPs, displayed different topical content, with plain packaging). Transcripts were analyzed using thematic analysis.

**Setting** Rural Menoufia, urban and semi-urban Cairo, Egypt.

**Participants** 90 waterpipe smokers and non-smokers, men and women, aged 18 years or older.

**Outcomes** Perceived potential effect on WTS uptake or cessation, probing factors related to PHW content and WTP design.

**Results** Participants in focus groups and in-depth interviews thought existing WT PHWs elicited affective responses, but found them unclear or unrealistic and thought the colourful packaging detracted from the warnings. In contrast, they thought novel and larger WT PHWs presented in plain packaging might prevent WTS initiation or trigger quit attempts. Participants regarded warnings featuring proximal health risks as most likely to be acceptable.

**Conclusions** Our exploratory study suggests larger WT PHWs featuring proximal risks and presented on plain WTPs could potentially deter experimentation with WT products among non-users and promote cessation among existing users.

## Strengths and limitations of this study

► This is the first qualitative study to explore plain packaging of waterpipe tobacco (WT) products in a country that has existing WT pictorial health warnings (PHWs).
► We provide novel insights from both non-smokers and smokers into potential policy-relevant outcomes, particularly uptake and cessation of WT smoking.
► Use of combined focus groups and in-depth interviews as qualitative methods offered rich understanding of perceptions related to WT labelling, with respect to which contents PHWs might feature, and how design of WT packs might be improved.
► Our sample of 90 individuals means we cannot generalise our findings; however, we included a variety of participants, and achieved data saturation.
► While we explored projected rather than actual responses to existing and novel WT PHWs with plain packaging, our findings could guide future experimental studies and assist policy-makers to improve WT regulations.

## INTRODUCTION

The introduction of flavoured tobacco and the lack of regulatory policies have seen waterpipe tobacco smoking (WTS) increase globally.[1 2] Misperceptions that WTS is a safe alternative to cigarette smoking may also have contributed to rising waterpipe tobacco (WT) use,[3] even though WTS causes respiratory illnesses, cardiovascular diseases and adverse perinatal outcomes.[4] These factors have helped WTS become more socially acceptable globally, especially among youth[5 6] and women.[7 8]

WT use has extended beyond the East, where it has been present for decades, and is increasingly popular in the West, where

WTS rates have reached 10% among some young adult populations in the USA and the UK.[9] [10] The WHO Eastern Mediterranean Region (EMR) remains home to the highest WTS rates worldwide[1] [11]; in some EMR countries, WTS has surpassed cigarette smoking in women and adolescents.[12] [13] Egypt has witnessed a rising trend in WT use; adolescent girls (3.4%)[14] and university students (12.2%)[15] report higher WTS rates than their older counterparts (0.3% in women[14] and 6.2% in men[16]), and rurally located Egyptian men smoke WT more (7.5%) than men living in urban regions (4.9%).[16] This global surge in WTS makes examining the perceived effectiveness of existing WT control policies important to inform a much needed WT regulatory framework.[17–19]

Applying health warnings to tobacco products can cost-effectively increase public awareness of smoking risks, increase the likelihood of quitting among smokers and deter smoking initiation among non-smokers.[20] These outcomes are mediated by several measures of effectiveness that have been organised, based on behavioural theories, within conceptual frameworks of health warning impact.[21–24] In line with this evidence, guidelines for implementing Article 11 of the WHO Framework Convention on Tobacco Control (WHO FCTC) call for on-pack pictorial health warnings (PHWs), and recommend plain packaging and increasing warning size.[25] Egypt, a signatory country to the WHO FCTC, has applied generic PHWs to waterpipe tobacco packs (WTPs).[26] Since 2008, a set of four PHWs has appeared on the bottom half of the front and back of WTPs; these warnings carry the quitline number and rotate every two years.[27] However, WTPs still depict colourful fruits and flavours in brand imagery.[27]

Several observational[28] and experimental studies[21] suggest plain packaging with larger PHWs[29] could more effectively reduce tobacco smoking through increasing warning salience, making the packaging and smoking less appealing, and reducing misperceptions about product harm,[30] [31] especially in non-smokers or non-established smokers.[32] Yet, while this evidence is encouraging, these studies have focused largely on cigarettes and we know little about how PHWs could reduce non-cigarette tobacco use, particularly WTS.[33]

To our knowledge, only a few studies have examined the impact of WT PHWs: two online surveys from Canada[34] and the USA,[35] three qualitative studies from the UK,[36] Egypt[37] and the EMR[38] and one recent Egyptian survey.[39] The two online surveys tested hypothetical warnings shown on computer screens rather than on WTPs and examined the effectiveness of text-only versus PHWs.[34] [35] Both studies found that PHWs had a modest impact on established waterpipe smokers.[34] [35] The UK qualitative study found that when warnings increased in size and packs became less branded, participants felt WTPs were less attractive and warnings were more impactful.[36] EMR study participants reported that PHWs improved respondents' knowledge about WTS health hazards.[38] These studies are important but were confined to waterpipe smokers[34–36] [38]; while it is important to examine how WT

PHWs might pertain to both smokers and non-smokers. The Egyptian qualitative study examined smokers' and non-smokers' responses to placement of PHWs on the waterpipe device. Participants reported this approach could potentially increase salience of WT PHWs, deter initiation of WTS and prompt non-established waterpipe smokers to quit.[37] The Egyptian survey reported that only half of 1048 waterpipe smoker and non-smoker participants thought that existing PHWs on WTPs were visible; they expressed varying views on the effectiveness of WT PHWs across several measures (such as salience, credibility, perceived harm, affective reactions).[39] However, this survey did not examine whether participants perceived existing WT PHWs effective in deterring uptake or quitting of WTS.

Given rising WTS rates in Egypt,[14–16] in 2015, the Tobacco Control Unit in the Egyptian Ministry of Health proposed amending PHW regulations and introducing plain packaging. Specifically, it recommended increasing PHW size to 80% of the pack surface, and removing colours and flavour imagery from tobacco packs. To provide preliminary insights into the potential effects of this approach, we used qualitative methods, which are particularly suited to exploring understudied areas.[40] [41] To inform WT labelling policy, we explored how participants perceived existing PHWs on WTPs, assessed how they interpreted a hypothetical scenario where WT was presented in plain packaging and featured enhanced PHWs and probed perceptions of how existing and novel PHWs would affect uptake or cessation of WTS.

## METHODS
### Design

The study comprised 10 focus group discussions and 10 in-depth interviews that took place in urban and semi-urban regions in Cairo, and a rural area in Menoufia governorate. We used both focus groups and in-depth interviews as complementary approaches; focus groups explored participants' interactions and whether and how consensus views evolved while in-depth interviews allowed detailed probing and deeper understanding of participants' views.[40] [41] Some sessions were conducted in WTS usage settings, such as cafes, where we observed social and cultural dynamics of WTS and assessed WT PHW's visibility to others.

Our conceptual framework drew on the theory of planned behaviour,[42] as outlined in IARC Handbooks of Cancer Prevention, Tobacco Control, Methods for Evaluating Tobacco Control Policies 2008.[24] We explored policy-relevant outcomes with respect to the perceived potential effect on WTS uptake and cessation, and probed factors related to PHW content and WTP design, including salience, affective reactions, perceived harm and credibility. We used Standards for Reporting Qualitative Research guidelines.[43]

## Sample

Our sample comprised men and women, 18 years of age or older, who lived in rural, urban and semiurban locations. We included self-identified waterpipe smokers (exclusive WT or dual users of WT and cigarettes) and non-smokers (non-users of any tobacco product), as we were interested in how warnings could influence WTS initiation as well as cessation. Participants were recruited using snowball sampling,[44] which enabled us to access female waterpipe smokers more easily and thus address calls for more research into this hard-to-reach group. We explained the study purpose to people who made contact, invited them to participate in a one-on-one interview or focus group and then set a meeting date and time. Participants did not discernibly differ by type of interview chosen.

In total, 90 individuals participated, including 80 in homogenous focus groups (with respect to age, gender, smoking status) with 6–8 individuals per group, and 10 in in-depth interviews (see online supplementary table 1). As WT use in Egypt is generally higher among men,[16] more men participated in our sessions. Online supplementary table 2 contains details of participants' demographic characteristics.

## Tools

### Interview guide

We developed the interview guide in Egyptian colloquial Arabic and incorporated qualitative measures used to assess tobacco labelling policies.[24] We pilot tested the interview guide for clarity and comprehensiveness, tested the appropriateness of our prompts and questions after the pilot sessions and made modifications following discussions with the research team. We used the same guide with focus groups and in-depth interviews and probed participants' experiences of WTS, their knowledge of WT PHWs, their views on the existing and novel PHWs on WTPs and their perceptions of placing PHWs on waterpipe devices (see online supplementary materials for interview guide). In this article, we focus on discussing PHWs on WTPs.

### Pictorial Health Warnings on Waterpipe Tobacco Packs

The existing PHWs depicted cancers of lung, throat, mouth and face, covered 50% of the lower surface of the front and back of the WTPs against a colourful background depicting fruit and flavour imagery. The warnings included pictures, generic text and the quitline number (figure 1A).

We adapted novel PHWs from a health warning database[45] and followed WHO FCTC recommendations for plain packaging[25] and WHO's publication on Evidence, Design and Implementation of Plain Packaging[46] building on the proposal of the Tobacco Control Unit of the Egyptian Ministry of Health; the PHW thus covered 80% of the upper surface of the front and back of the WTP against a dark uniform plain background not depicting any fruit or flavour imagery, with the remaining 20% depicting only the brand name in standardised font. The novel PHWs included pictures, text and the quitline number. Dark plain packs are perceived as more harmful[46] and

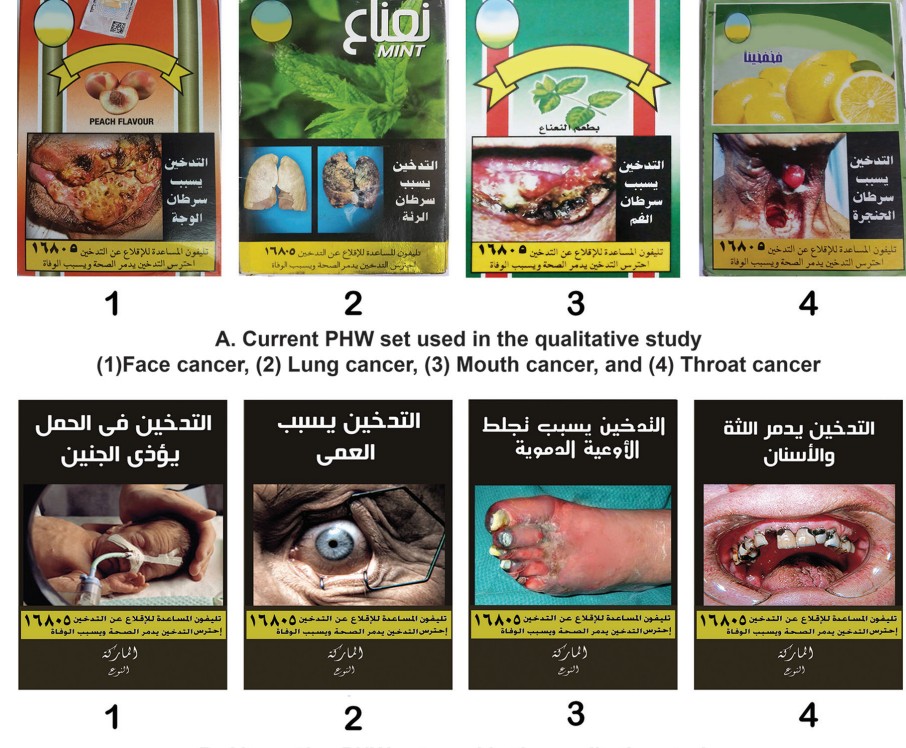

**A. Current PHW set used in the qualitative study**
(1)Face cancer, (2) Lung cancer, (3) Mouth cancer, and (4) Throat cancer

**B. Alternative PHW set used in the qualitative study**
(1)Fetal harm, (2) Blindness, (3) Blood vessel clotting, (4) Tooth and gum decay

**Figure 1**   Pictorial health warnings (PHWs) used in this study.

culturally as more negative (when compared with the bright background colours of the existing WTPs) and we therefore used a drab dark brown colour (similar to that used in Australian packaging) on novel WTPs. Feedback from pilot testing indicated that the dark background colour contrasted well with the white colour of the textual message and the yellow background of the quitline number, making both more clear and salient. We applied newly designed PHWs to used WTPs to promote authenticity. We pilot tested health warning messages with corresponding images for clarity and comprehension: 'Smoking kills,' 'Smoking causes lung cancer', 'Smoking causes clotting of blood vessels', 'Smoking causes blindness', 'Smoking during pregnancy harms fetus' and 'Don't let your children inhale your smoke'. The warnings used were selected following discussions among the research team and feedback from pilot sessions, and reflected the best available evidence on WTS health outcomes.[47] WTS share common harms with those caused by cigarette smoking.[4] We sought topical content showing less severe harms than those depicted in existing PHWs. The four novel warnings selected included the effect of smoking during pregnancy on the fetus, effects of peripheral vascular diseases affecting the feet and eye and effects on teeth and gums (figure 1B). Although it was important to adapt the textual message to be waterpipe-specific, we did not test this in the study reported here; this was assessed separately in another study of our research project.

## Data collection

Data were collected from October 2015 to February 2016 at the Faculty of Medicine, Ain Shams University in Cairo (five focus groups and five in-depth interviews), and at participants' homes or in cafes for those in rural and semiurban areas (five focus groups and five in-depth interviews). All participants received an information letter explaining the study and were asked to provide verbal consent prior to each discussion or interview commencing. Participants were advised their data and identity would be confidential, and told they could withdraw from the study at any time.

Each focus group or interview was moderated by two of the coauthors (AM, WS, ME and WMH) and audio-recorded; each session was about one hour long. The facilitator and note taker regularly switched roles to promote reflection, and the wider team critically reflected on the interviews during team meetings. The facilitators followed standard procedures when discussing the interview guide topics and introduced the PHW stimuli when the relevant topic was opened for discussion. No further sessions were scheduled once data saturation had been reached, that is, no new themes were being generated during the discussions.[48]

## Analysis

Two authors independently transcribed verbatim the recorded sessions in their original language then translated these into English before comparing the two transcripts to ensure inclusivity and accuracy (WMH, HM); a third author (AM) back translated the transcripts independently for validity purposes; any discrepancies were resolved through discussion. Considering the identical aims and topics explored, data from focus groups and interviews were analysed together. We analysed the data using a thematic approach.[49 50] We coded transcripts as the study progressed using a three-phase process that began by organising ideas in relation to the research questions, then involved independent reviews of transcripts to identify preliminary themes and create an initial coding list.[51] We finally independently refined this list (AM, AA), added new codes where appropriate and developed broader themes, that one author (HM) then reviewed across all cases. We resolved minor inconsistencies during discussion sessions and after extensive reviews of transcripts before we finalised the themes and subthemes (see online supplementary table 3). In this article, we focus on policy-relevant outcomes relevant to PHWs on WTPs as described above in Methods 'Design'.

## Patient and public involvement

Patient and public were not involved in the development of the research question and outcome measures, the design, recruitment and conduct of the study. The results of this study will be disseminated to study participants via newsletters and social media outlets.

## RESULTS

The 90 participants in focus groups and interviews comprised more men (72.2%) than women (27.8%) and participants' mean age was 33.4±11.6 (see online supplementary table 2). We identified the overall themes: warning label content and pack design features, and discuss these in relation to WTS uptake and cessation comparing throughout existing and novel sets. During analysis of the transcripts, we did not detect differences between the focus group and individual interview data. Therefore, we report below results from both focus groups and individual interviews. We cite exemplar quotations below and provide a more detailed set of quotations in online supplementary table 4, where we also indicate the gender, age group, smoking status, location of participants and source of quotations.

Most participants were aware of warning labels on WTPs and reported seeing these when purchasing or preparing their tobacco. However, those using waterpipes in cafes were less likely to see WTPs as WT was prepared out of their sight 'I was downtown and we saw smokers in cafes but shisha is always served ready…I never saw the packs' (Female non-smoker, >25 years old, semiurban, FGD).

## Warning label content: perceived likely effect of existing and novel PHWs

Participants who were aware of existing WTP warnings recalled these as disturbing; however, several felt these warnings had limited impact. Many recalled the lung

cancer PHW and the text 'smoking is hazardous to health and causes death' as the most believable warning, yet also the least impactful, because of wearout: '…the other warnings like the lung cancer one…people got used to them after a while' (Male smoker, >25 years old, rural, IDI). Participants found the existing warnings, which all featured cancer-related harms, frightening and disgusting, and non-smokers, particularly, avoided looking at them. Nonetheless, several questioned the harms existing PHWs featured and saw these as exaggerated: 'I don't want to look at it from near or far…they want to send us a message that it is harmful but in an awful and overstating way' (Female smoker, <25 years old, urban, FGD).

Some also found existing warnings difficult to understand: 'It looks like a bad thing but it is not clear what it is' (Female non-smoker, >25 years old, semiurban, FGD). These participants saw existing PHWs as unconvincing and exaggerated as they had not seen such conditions in real life and sometimes knew people who had smoked for many years with apparent impunity: 'I have seen the warning on the packs but I have never seen anything such as that in real life…I know a person who has been smoking since the eighties and nothing happened to him' (Male smoker, >25 years old, rural, FGD).

Participants often denied risks associated with occasional smoking; one exempted himself from harm on the grounds that he did not smoke heavily and had seen no direct evidence of the harms presented in PHWs: 'I don't think I can be affected by smoking, because I don't smoke heavily; only once a day, besides…we have never seen the conditions in the warnings in real life' (Male smoker, >25 years old, rural, IDI). Several thought harms would not occur until they were older, and believed that, if and when they experienced these, they could quit.

By contrast, most participants favoured the new PHWs, which they first viewed during the interview sessions. They found these clearer, more understandable and realistic than existing PHWs, and more likely to capture their attention. Participants also commented that the new PHWs were easily understood even without text and thus likely to be effective among people with varying literacy: 'This one is more realistic for its purpose and understandable; even without any text…it is a more convincing warning' (Female non-smoker, >25 years old, semiurban, FGD).

Specifically, participants found the new warnings featuring more immediate, proximal risks had the strongest perceived impact, particularly those showing harmful effects on teeth. Female participants also felt strongly affected by the PHW illustrating harm to unborn babies, which forced them to confront the harm they imposed on others: 'The baby warning is effective because it is not about me anymore…This is something way more important than me…I fear for my kid more than I fear for myself' (Female non-smoker, >25 years old, semiurban, FGD). However, some continued to see images showing potential harms they may experience as overstated and unlikely to happen in the near future: 'This didn't happen before to anyone (foot warning)…we're still young… we won't get this' (Female smoker, <25 years old, urban, FGD).

Although participants saw the new PHWs as more effective, they suggested improvements to the warning content. In line with earlier comments about perceived exaggeration, they sought greater credibility: 'It is very important that you convince me…put something there that I'll believe' (Male smoker, >25 years old, semiurban, IDI). Some suggested presenting testimonials: 'Show smokers live people who were damaged because of shisha smoking and others who quit and improved' (Male non-smoker, >25 years old, rural, FGD), and others recommended illustrating the effects WTS has on women by featuring relevant cancers and congenital diseases: 'Direct warnings addressing women like cancer of the breast or the uterus or congenital anomalies to the fetus' (Female non-smoker, >25 years old, semiurban, FGD). Some also thought PHWs targeting women could encourage them to persuade their partners to quit.

Several participants thought PHWs should target youth before they start smoking and suggested warnings showing how WTS smokers' social relationships (eg, sexual dysfunction warnings) would have high impact: 'We want a real effect that already happened…like for example the side effects on sexual functioning…that will definitely affect smokers' (Male non-smoker, >25 years old, rural, FGD).

In addition, participants also suggested printing text warnings including details of the hazardous ingredients of smoke, with more information on health risks and cessation options inserted on the inside or outside of the WTP, together with external PHWs would promote cessation behaviour, and enhance the impact of novel warnings.

## Design features of WTPs: perceived likely effect of existing and novel PHWs

As well as responding to the different warning content, participants also noted that the different design elements used in the novel PHWs had improved the impact these had. Several commented on how bright colours and fruit imagery deflected attention away from PHWs and promoted experimentation: 'The peach drawing is appetising…(and) drawing attention away from the warning… better put the warning on the top! The pack should be dark…this colour is very bright…just like bonbon packs' (Female non-smoker, >25 years old, semiurban, FGD). They found it difficult to associate fruit flavours with harm and thought the images invited trial: 'The pictures and the smell of fruits make a passerby want to try them all' (Female non-smoker, >25 years old, semiurban, IDI).

By contrast, participants thought the plain background, contrasting colours, absence of fruit and flavour images and larger warning images shown on the proposed new PHWs increased impact, reduced distraction and encouraged participants to look more closely at the pack. One noted: 'Here the picture is bigger and the text has a clear

message…together with this dark colour…it makes me focus only on the warning…all this makes it more effective' (Male smoker, <25 years old, urban, FGD). Together, the altered content and enhanced design attributes increased participants' perceptions of the impact the novel PHWs would have.

Overall, most waterpipe smoker and non-smoker participants thought PHWs would deter non-smokers from trying WTS but were less optimistic about the effects on smokers. One noted: 'If we (non-smokers) lusted to smoke it and saw pictures like these…we won't smoke, but those who do actually smoke already would be indifferent' (Male non-smoker, >25 years old, urban, FGD). Smokers themselves also felt PHWs had less effect because they had become accustomed to seeing the images: 'I used to think about the hazards a lot when the pictures first appeared…then I got used to them…I don't pay them attention anymore' (Male smoker, <25 years old, semiurban, IDI). Others reported using stickers to obscure PHWs or avoiding packs with PHWs they found particularly confronting: 'I avoid buying the picture of the tongue in particular (referring to the mouth cancer warning)' (Male smoker, <25 years old, semiurban, FGD).

As with the current PHWs, most waterpipe smoker and non-smoker participants thought the proposed PHWs would have a stronger effect on non-smokers than on long-term smokers, though some indicated they would avoid some warnings and may reduce their WTS: 'If I go to buy moassel (waterpipe tobacco) and found this pack, I'll go to another shop to buy another one…if I don't find a picture that makes me comfortable…I won't smoke that day…but if they're all like this…I guess I'll try to quit… or…at least decrease my habit' (Male smoker, >25 years old, rural, IDI). In general, participants thought that the new PHW set had greater potential to deter WTS, especially among new smokers: 'If I'll smoke and saw it…for sure I won't smoke at the moment…it's disgusting' (Male non-smoker, <25 years old, rural, FGD).

## DISCUSSION

Our qualitative study found that participants privileged the short-term benefits they received over the longer-term risks they faced. They saw existing PHWs as less likely to influence long-term smokers,[52] especially if they had not experienced any health effects. This finding is consistent with other studies that found age is negatively associated with perceived risks of smoking and with attention to either graphic or text warnings on tobacco packages.[53]

However, the existing and novel PHWs tested appeared more likely to influence non-smokers and less-experienced smokers by creating awareness of the health risks WTS presents. These findings are in line with previous research[32] and address calls for research into the effects of tobacco packaging on smoking uptake.[54] We also provide preliminary evidence that presenting WT in plain packaging could deter non-smokers from experimenting WTS. The existing and the novel PHWs differed in three

main ways: the topical imagery content, the size of the warning and the pack design. We explain below how policy-makers should consider these three elements when adopting or amending regulations for PHWs on WTPs.

Although existing PHWs induced strong negative emotional reactions, several participants viewed these warnings as exaggerated and felt the health risk depicted was unlikely to occur. While PHWs that arouse fear may increase risk perceptions, they did not necessarily promote greater message acceptance.[55–57] Our findings also show a complex relationship between the emotional response elicited and the salience and perceived impact of a message. The PHW depicting oral harm was minimally disturbing, yet participants saw it as the strongest and the most salient warning; by contrast, participants regarded the confronting vascular harm PHW as less effective because the condition was less salient and seen as more distal. Our findings thus support earlier studies analysing the relationship between risk perception, believability and temporal distance.[58]

Young adults and women found gender-specific and age-specific messages more persuasive,[59] suggesting strategies targeting these demographic groups are crucially important.[60] For example, messages about negative health effects from passive smoking, sexual dysfunction and intimacy, miscarriages, and harmful effects on infants and children were considered more persuasive among younger adults and women than other messages.

Another plausible approach suggested by participants was to develop a visible and sensible message in both text and graphic formats that target waterpipe smokers with detailed cessation information inserted on the WTP. Combined text and graphic warnings elicit adverse reactions to smoking among non-smokers and smokers,[61] especially on the sustainability of quit behaviour.[62] In line with these suggestions, the design of novel warnings helped make the cessation quitline more prominent by the contrasting dark plain background.

We also found specific design elements that could inform future PHW development. Some participants thought plain packs increased the salience and effectiveness of the warnings, and reduced the appeal of the packaging and misperceptions of harm. These findings are consistent with the literature on plain packaging.[46 54] Evidence that flavoured cigarettes reduce harm perceptions appears relevant to WT[63 64] as participants noted that images of fruits or appealing flavours attracted them to WTS and deflected their attention from warning information. Furthermore, image clarity and size enhanced acceptance and risk perception among participants, a finding also reported by Jawad and colleagues.[36]

### Strengths and limitations

To our knowledge, this is the first qualitative study to assess smokers' and non-smokers' awareness and acceptance of currently used and novel WT PHWs with plain packaging. Our novel warnings simulated how PHWs appear (or could appear) on packs and devices; we explored

projected rather than real-life responses. Experimental work could estimate the likely impact of our new label designs, including more targeted PHWs, on waterpipe smokers' risk perceptions, attitudes and likely cessation responses. Future research could also develop a more comprehensive analysis of factors motivating and reinforcing WTS uptake. Given our findings that PHWs have the potential to reduce WTS initiation, it is also important to test whether the PHW themes we developed could be used in wider health promotion campaigns to reduce the appeal and perceived acceptability of WTS. Future research could explore how WT warnings featuring waterpipe-specific messages would affect awareness and perceptions of WTS.

Our small sample means we cannot generalise our findings, though we note a sample of 90 individuals is still substantial, and saturation had been reached in the responses received. Despite these limitations, our study provides novel insights into the factors supporting WTS uptake, suggests themes PHWs could feature and outlines how PHWs' format could be improved. We recognise this work programme faces challenges, given the limited resources and capacity in low/middle-income countries; nonetheless, our findings represent an important step in supporting a comprehensive regulatory framework that reduces WTS and the harm this form of tobacco use causes.

## CONCLUSIONS

This exploratory study suggests that PHWs on WTPs have the potential to reduce uptake and cue quit attempts but might be more effective if PHWs used more impactful designs. Specifically, we provide preliminary evidence that enhanced PHWs using contrasting background colours and plain packaging, offering no association to fruits or flavours, targeting age and gender and displaying proximal health risks, might enhance both warning impact and risk perception. These alternative designs could be further developed and tested in other studies. The findings offer policy-makers designing and implementing health warnings on WT products clearer evidence on which to base their decisions.

**Author affiliations**
[1]Department of Community, Environmental, and Occupational Medicine, Faculty of Medicine, Ain Shams University, Cairo, Egypt
[2]School of Pharmacy, University of Waterloo, Ontario, Canada
[3]Department of Psychiatry Medicine, Faculty of Medicine, Ain Shams University, Cairo, Egypt
[4]Egyptian Tobacco Control Coalition, Cairo, Egypt
[5]Egypt Health Foundation, Cairo, Egypt
[6]Tobacco Control Unit, Ministry of Health, Cairo, Egypt
[7]Departments of Public Health and Marketing, University of Otago, Otago, New Zealand

**Acknowledgements** We thank the participants of this research and Joe Petrik for copyediting a previous version of this article. The pictorial health warnings presented in this study were used after signing a written partnership agreement between the Tobacco Control Unit in the Egyptian Ministry of Health and the research team.

**Contributors** AM conceptualized and designed the study, developed its tools and wrote this article. AM, ME, WS and WMH carried out the study. SL provided approval for using pictorial health warnings. WMH and HM transcribed the data. AA, HM, AM and JH analysed the data. AM, HM, AA and JH further developed this article. All authors critically revised contents and provided final approval on the submitted manuscript.

**Funding** This work was carried out with the aid of a grant from the International Development Research Center, Ottawa, Canada through the American University of Beirut, the Tobacco Control Research Group, to study waterpipe tobacco smoking prevention and intervention programs in the region, as a part of the project "Shaping Research for Health in the Arab World: A Systems and Network Approach to Advance Knowledge, Inform Policy, and Promote Public Health" (Grant 106981-001).

**Competing interests** None declared.

**Patient consent** Not required.

**Ethics approval** This study was approved by the Research Ethics Committee of Faculty of Medicine, Ain Shams University (FMASU R 10/2015).

**Provenance and peer review** Not commissioned; externally peer reviewed.

**Data sharing statement** No additional data are available.

**Author note** Checklist for reporting guidelines: the authors used SRQR guidelines for reporting qualitative research.

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
