## [Reviewer comments · BMJ Open]

ARTICLE DETAILS

TITLE (PROVISIONAL)	Plain Packaging of Waterpipe Tobacco? A Qualitative Analysis Exploring Waterpipe Smokers' and Non-smokers' Responses to Enhanced versus Existing Pictorial Health Warnings in Egypt
AUTHORS	Mostafa, Aya; Mohammed, Heba Tallah; Mohamed, Wafaa; Elhabiby, Mahmoud; Safwat, Wael; Labib, Sahar; Aboul Fotouh, Aisha; Hoek, Janet

VERSION 1 – REVIEW

REVIEWER	Mohammed Jawad Imperial College London, United Kingdom
REVIEW RETURNED	24-Apr-2018

GENERAL COMMENTS	This large qualitative study conducted in Egypt aimed to assess existing and potential future health warning labels on waterpipe tobacco. The study is important due to the increase in waterpipe tobacco smoking in the Egyptian population (and worldwide) and the need for policy research to guide waterpipe tobacco control. To the best of knowledge this is among the first of studies to report such research. The introduction is well-written and defines the problem and its context. Statements are substantiated with up-to-date literature. I have no suggestions for improvement here. The methods are also well-described and are grounded in a conceptual framework. The authors sampled from multiple areas and considered homogeneity (age, sex, smoking status) in the construction of their focus groups. This is a positive step towards clearly understanding and answering the research question at hand. The authors also followed an interview guide and conducted a pilot study, which is expected for a high-quality qualitative research piece. The enhanced health warnings followed expected packaging and labeling requirements from the Egyptian Ministry of Health which makes this study extremely policy-relevant. Some justification is needed for the snowball sampling - was this really necessary given waterpipe tobacco use is common in Egypt? Were there practical or epidemiological considerations for this? Secondly, who translated the transcripts from Arabic to English, and how do we know that the integrity of the statements was maintained during the translation process? (e.g. were they back translated independently for validity purposes?) The results are clearly written and additional information is provided in the supplementary material, which makes this qualitative study quite rich. A few comments for consideration:
--

	- Were there any differences in results from focus groups and interviews? Although the two approaches are similar, there still remains nuanced differences that may bring out different opinions and perceptions. - On how many variables/features did the two sets of health warnings differ? For example, were the enhanced warnings both larger in size, and have a different message content, and a different image? I ask because the existing health warnings were described as unrealistic and scary - presumably this is the message/image - whereas the larger ones were described more realistic. A policymakers may see these results and suggest that changing message content is more appropriate than increasing size. The discussion has good content and is insightful, but it was quite lengthy and could do with being more concise. The fact that the limitations section of the paper is in a supplementary material is testimony to this.
--	---

REVIEWER	Ilze Bogdanovica University of Nottingham, UK
REVIEW RETURNED	10-Jun-2018

GENERAL COMMENTS	Introduction:  • First sentence needs to be supported by evidence if authors are suggesting global increase in waterpipe usage • “These factors have helped WTS become more socially acceptable, especially among youth and women”- needs to be specified whether this refers to local or national context • Page 8 2nd paragraph- when authors make reference to rising WTS rates in Egypt specific figures should be reported. • Authors need to clarify in intro that their study is about hypothetical scenario and not the actual implemented legislation in Egypt Methods:  • The sample includes participants aged 18+ yet one of the aims is to understand how warnings influence initiation. Is this the right age group for this as tobacco use typically is started before age of 18? • Authors need to justify use of snowball sampling • How did researchers decide on the design of plain packaging, for example, how was “a dark uniform plain background (page 11)” selected? More detailed description on choosing hypothetical scenario would be beneficial Discussion:  • Authors should be more specific on what this study adds • What is the rationale for inclusion of strengths and limitations as a supplementary document? It is an integral part of discussion and should be presented as such.
--

VERSION 1 – AUTHOR RESPONSE

Reviewer: 1

This large qualitative study conducted in Egypt aimed to assess existing and potential future health warning labels on waterpipe tobacco. The study is important due to the increase in waterpipe tobacco smoking in the Egyptian population (and worldwide) and the need for policy research to guide

waterpipe tobacco control. To the best of knowledge this is among the first of studies to report such research.

The introduction is well-written and defines the problem and its context. Statements are substantiated with up-to-date literature. I have no suggestions for improvement here.

The methods are also well-described and are grounded in a conceptual framework. The authors sampled from multiple areas and considered homogeneity (age, sex, smoking status) in the construction of their focus groups. This is a positive step towards clearly understanding and answering the research question at hand. The authors also followed an interview guide and conducted a pilot study, which is expected for a high-quality qualitative research piece. The enhanced health warnings followed expected packaging and labeling requirements from the Egyptian Ministry of Health which makes this study extremely policy-relevant.

Some justification is needed for the snowball sampling - was this really necessary given waterpipe tobacco use is common in Egypt? Were there practical or epidemiological considerations for this?

Authors' response: We chose this approach as female waterpipe smokers were difficult to reach. Adult female waterpipe tobacco smoking has been reported to be less than 1%; it was expected to find a minimal number of female waterpipe smokers. Some tobacco control experts considered there has been under-reporting of female waterpipe tobacco smoking. We assumed they did not want their WTS status to be known. Hence, we used the same sampling approach for males and females. We have clarified this information in the revised methods: "Participants were recruited using snowball sampling,⁴⁴ which enabled us to access female waterpipe smokers more easily and thus address calls for more research into this hard-to-reach group."

Secondly, who translated the transcripts from Arabic to English, and how do we know that the integrity of the statements was maintained during the translation process? (e.g. were they back translated independently for validity purposes?)

Authors' response: To ensure that the integrity of the statements was maintained during the translation process, all focus groups and interviews were transcribed in Arabic (in order not to lose the meaning and to preserve the fidelity of language-specific constructs and avoid errors that can occur when translation occurs at the point of transcription) then translated into English by two bilingual researchers and independently validated by a third researcher. We have clarified this point under "Analysis": "Two authors independently transcribed verbatim the recorded sessions in their original language then translated these into English before comparing the two transcripts to ensure inclusivity and accuracy (WH, HM); a third author (AM) back translated the transcripts independently for validity purposes; any discrepancies were resolved through discussion."

The results are clearly written and additional information is provided in the supplementary material, which makes this qualitative study quite rich. A few comments for consideration:

- Were there any differences in results from focus groups and interviews? Although the two approaches are similar, there still remains nuanced differences that may bring out different opinions and perceptions.

Authors' response: During analysis of the transcripts, we did not detect differences between the focus group and individual interview data. As a result, we analysed these data jointly and drew on the combined data to develop the themes we present in the manuscript. We included quotations from both the focus groups and individual interviews in the results section and in the supplementary file, and note that we are drawing on the combined data: "...we report below results from both focus

groups and individual interviews.”. We have now made that point clearer in the revised results: “During analysis of the transcripts, we did not detect differences between the focus group and individual interview data. Therefore, we report below results from both focus groups and individual interviews.”

For example, the following quote from individual interviews that we presented in the results: “This didn’t happen before to anyone (foot warning)...we’re still young...we won’t get this” (Female smoker, <25y, urban) was similar to those from the focus groups: “I just don’t believe it...nobody reaches this stage” (Female smoker, <25y, urban) and “The pictures are not realistic, we never see such things in real life” (Male non-smoker, >25y, semi-urban).

We have added the source to each of the quotations that we presented in both the results and in the supplementary file (FGD for focus groups and IDI for individual interviews) and have noted that in the revised results: “We cite exemplar quotations below and provide a more detailed set of quotations in Supplementary Table 4, where we also indicate the gender, age group, smoking status, location of participants and source of quotations.”

- On how many variables/features did the two sets of health warnings differ? For example, were the enhanced warnings both larger in size, and have a different message content, and a different image? I ask because the existing health warnings were described as unrealistic and scary - presumably this is the message/image - whereas the larger ones were described more realistic. A policymakers may see these results and suggest that changing message content is more appropriate than increasing size.

Authors’ response: We have clarified this information in the revised discussion section: “The existing and the novel PHWs differed in three ways: the topical imagery content, the size of the warning, and the pack design. We explain below how policymakers should consider these three elements when adopting or amending regulations for PHWs on WTPs...”.

The discussion has good content and is insightful, but it was quite lengthy and could do with being more concise. The fact that the limitations section of the paper is in a supplementary material is testimony to this.

Authors’ response: We have revised the discussion to be more concise and have included the limitations section therein.

Reviewer: 2

Introduction:

- First sentence needs to be supported by evidence if authors are suggesting global increase in waterpipe usage

Authors’ response: The focus of reference no.1 cited was the global increase in waterpipe usage and reasons for this increase. We have added another citation of a more recent systematic review as supporting evidence of increasing waterpipe usage trends (reference no.2): “Jawad M, Charide R, Waziry R, et al. The prevalence and trends of waterpipe tobacco smoking: A systematic review. PLoS ONE 2018; 13(2): e0192191. doi.org/10.1371/journal.pone.0192191.”

- “These factors have helped WTS become more socially acceptable, especially among youth and women”- needs to be specified whether this refers to local or national context

Authors’ response: We have made this point clearer: “These factors have helped WTS become more

socially acceptable globally, especially among youth^{5,6} and women.^{7,8} The context of the first paragraph is the global situation. The references cited [5-8] provide support for our claim that global acceptability of WTS has increased.

- Page 8 2nd paragraph- when authors make reference to rising WTS rates in Egypt specific figures should be reported.

Authors' response: We have amended the text to make this point more explicit: "Egypt has witnessed a rising trend in WT use; adolescent girls (3.4%)¹⁴ and university students (12.2%)¹⁵ report higher WTS rates than their older counterparts (0.3% in women¹⁴ and 6.2% in men¹⁶), and rurally-located Egyptian males smoke WT more (7.5%) than men living in urban regions (4.9%).¹⁶". We reported specific figures in the second paragraph of the introduction to document the rising WTS rates in Egypt. The references cited [14-16] refer to different time points and thus demonstrate the increasing trend.

- Authors need to clarify in intro that their study is about hypothetical scenario and not the actual implemented legislation in Egypt

Authors' response: We have revised the text to make this point clearer: "To provide preliminary insights into the potential effects of this approach...assessed how they interpreted a hypothetical scenario where WT was presented in plain packaging and featured enhanced PHWs...". We described briefly the actual implemented legislation in page 7 first paragraph. We also stated in page 8 second paragraph that this was a proposal (i.e. not actual implemented legislation).

Methods:

- The sample includes participants aged 18+ yet one of the aims is to understand how warnings influence initiation. Is this the right age group for this as tobacco use typically is started before age of 18?

Authors' response: We note that adults are still susceptible for waterpipe tobacco use/initiation, as the mean age of initiating WTS among participants in a parallel survey of our research project was 18.3±3.5 (18.0±3.2 for males and 21.9±4.5 for females) and ranged from 11 to 40 years old (data unpublished yet).

- Authors need to justify use of snowball sampling

Authors' response: Please see our response to the same comment by Reviewer 1.

- How did researchers decide on the design of plain packaging, for example, how was "a dark uniform plain background (page 11)" selected? More detailed description on choosing hypothetical scenario would be beneficial

Authors' response: Thank you for raising this important point. We described the basis of the pack design in the hypothetical scenario and have explicitly mentioned the references that formed the basis for selection of the novel pack design: "We adapted novel PHWs...and followed WHO FCTC recommendations for plain packaging²⁵ and WHO's publication on Evidence, Design and Implementation of Plain Packaging 46 ...".

- Reference no.46 (WHO's publication on Evidence, Design and Implementation of Plain Packaging) states that: "Perceptions of the harmfulness of tobacco products were found to depend primarily on the colour of packaging, with darker plain packs seen as more harmful." (page 15 of reference 46).
- In reference no.46, Figure 2 (page 27 of reference 46) demonstrates the requirements of the

Australian regulations for cigarette plain packs, where the colour of the pack surface is Pantone 448C (a drab dark brown). We used a similar colour on our stimulus material.

- Reference no.25 states that: "Parties should select contrasting colours for the background of the text in order to enhance noticeability and maximize the legibility of text-based elements of health warnings and messages.". We found that the dark background colour contrasted well with both the white color of the textual message and the yellow colored background of the quitline number that is shown on the novel waterpipe packs beneath the pictorials.

-The dark color is also perceived culturally as negative/sad/repulsive in contrast to the inviting bright orange, and green colors that were used as background for figurative signs of fruits and flavors in the existing PHWs.

-Moreover, in pilot testing of the packs participants commented that it invited more scrutiny to the content of the warning and made the warning message more clear and visually salient.

Accordingly, we have now extended the description on how the background of plain packs was selected: "Dark plain packs are perceived as more harmful⁴⁶ and culturally as more negative (when compared with the bright background colors of the existing WTPs) and we therefore used a drab dark brown colour (similar to that used in Australian packaging) on novel WTPs. Feedback from pilot testing indicated that the dark background colour contrasted well with the white color of the textual message and the yellow background of the quitline number, making both more clear and visually salient. ". References no.25 and 46 are cited for referring to more details of the recommended background for plain tobacco packs if needed.

Discussion:

- Authors should be more specific on what this study adds

Authors' response: We have revised the conclusions to make what this study adds more explicit: "This exploratory study suggests that PHWs on WTPs have the potential to reduce uptake and cue quit attempts but might be more effective if PHWs used more impactful designs. Specifically, we provide preliminary evidence that enhanced PHWs using contrasting background colours and plain packaging, offering no association to fruits or flavours, targeting age and gender, and displaying proximal health risks, might enhance both warning impact and risk perception. These alternative designs could be further developed and tested in other studies. The findings offer policymakers designing and implementing health warnings on WT products clearer evidence on which to base their decisions."

- What is the rationale for inclusion of strengths and limitations as a supplementary document? It is an integral part of discussion and should be presented as such.

Authors' response: We have now included comments about the study's strengths and limitations to the revised discussion section.

VERSION 2 – REVIEW

REVIEWER	Mohammed Jawad Imperial College London
REVIEW RETURNED	20-Jul-2018
GENERAL COMMENTS	The authors have adequately responded to my comments and I have no further suggestions to improve the manuscript. I wish them all the best with their ongoing research in this area.